# Tumor-Associated Tractography Derived from High-Angular-Resolution Q-Space MRI May Predict Patterns of Cellular Invasion in Glioblastoma [note 1]

**DOI:** 10.3390/cancers16213669

**Published:** 2024-10-30

**Authors:** Owen P. Leary, John P. Zepecki, Mattia Pizzagalli, Steven A. Toms, David D. Liu, Yusuke Suita, Yao Ding, Jihong Wang, Renjie He, Caroline Chung, Clifton D. Fuller, Jerrold L. Boxerman, Nikos Tapinos, Richard J. Gilbert

**Affiliations:** 1Department of Neurosurgery, Warren Alpert Medical School, Brown University, 222 Richmond St., Providence, RI 02903, USAnikos_tapinos@brown.edu (N.T.); 2Department of Neurosurgery, Brigham and Women’s Hospital, 75 Francis St., Boston, MA 02115, USA; 3Seattle Children’s Hospital, 4800 Sand Point Way NE, Seattle, WA 98105, USA; 4Department of Radiation Physics, University of Texas MD Anderson Cancer Center, 1515 Holcombe Blvd., Houston, TX 77030, USA; 5Department of Radiation Oncology, University of Texas MD Anderson Cancer Center, 1515 Holcombe Blvd., Houston, TX 77030, USAcdfuller@mdanderson.org (C.D.F.); 6Department of Diagnostic Imaging, Warren Alpert Medical School, Brown University, 222 Richmond St., Providence, RI 02903, USA; jboxerman@lifespan.org; 7Roger Williams Medical Center, 825 Chalkstone Avenue, Providence, RI 02903, USA; rjgilbert12@gmail.com

**Keywords:** glioblastoma, q-space imaging, diffusion, transcriptomics, patient-derived xenograft, cancer stem cells, tumor invasion

## Abstract

While tumor cell invasion beyond the surgically resectable “margin” of glioblastoma is thought to be associated with nearly 100% of recurrences and poor survival, no reliable methods exist for mapping the location of invading tumor cells within the human brain. Building on prior work demonstrating the ability of Q-space magnetic resonance imaging (QSI) to highlight structural alterations in tissue architecture, we hypothesized that using this imaging method to construct tumor-associated tractography might identify tumor-specific structures that underlie cellular invasion. Our results demonstrate that such tractography patterns can be observed in a tumor-specific manner, and provide preliminary evidence that those patterns may colocalize with invading tumor cells, and metrics derived from them may be associated with patient survival time.

## 1. Introduction

Glioblastoma is the most common and most lethal primary malignant tumor of the brain [1]. A noted attribute of these tumors is their resistance to virtually all forms of therapy [2,3,4,5], which may be related to underlying cellular phenotypic plasticity and a propensity to invade normal brain tissue [6,7,8,9]. Moreover, this biological attribute may be associated with distributed patterns of proliferation and migration under the influence of local force, nutrients, and growth factors [10,11]. Large scale analyses of glioblastoma have confirmed that considerable genomic and epigenetic heterogeneity between tumors exists [12,13,14,15], in addition to significant cellular heterogeneity within tumors [6,16,17,18], which together may promote the ability of tumor cells to evade treatment. Furthermore, glioma cells have been shown to closely associate with neuronal components within the brain, including the formation of functional electrical and chemical connections with neighboring cells [19,20] as well as migration along axonal structures [21,22,23]. It is likely that cellular and molecular promoters for glioblastoma growth and invasion are complex and affect the biological behavior of the tumor through spatially distributed regulatory pathways in the neighboring microenvironment [24]. Some evidence also suggests that subsets of glial cells within the tumor, particularly stem cell populations which may also serve as tumor initiating cells in malignant gliomas, function much like migrating neural and glial cells in the perinatal brain [7,9,25,26,27,28]. These cells may recapitulate a developmental phenotype which subserves robust migratory interactions with white matter tracts intersecting the tumor [21].

We postulated that the directional migration of glioblastoma cells into the brain parenchyma follows trajectories dependent on programmed interactions with axons and non-malignant glial cells. However, despite the importance of discerning structural mechanisms for tumor migration [29], no methods exist to reliably map the distribution and progression of these invading cells after disease diagnosis. Diffusion-weighted magnetic resonance imaging (dwMRI) characteristically depicts tissue organization based on the orientation of restricted molecular displacement [30,31]. More broadly, dwMRI is sensitive to the displacement of water molecules in the presence of an oriented magnetic field, whether or not those movements are restricted by compartmentation due to membranes or other cellular, subcellular, or extracellular structures. dwMRI methods have been applied for mapping functionally important white matter tracts of the brain based on preoperative MRI scans, which can then be registered to patient anatomy intraoperatively using surgical navigation units during tumor resection [32,33,34,35]. Diffusional metrics of potential diagnostic and prognostic utility have also been identified [36,37,38,39,40,41], including preliminary assessment of tractography-derived metrics [39,40,42]. Despite these current applications of dwMRI in neuro-oncology, these methods have not yet been rigorously investigated as a means of directly mapping tumor cell invasion.

In this study, we applied Q-space magnetic resonance imaging (QSI), a dwMRI method capable of generating high resolution maps of fiber alignment through the analysis of sub-voxel diffusivity [43,44], in patients with glioblastoma. We hypothesized that patterns of tumor-associated tractography may be able to predict the location of invading tumor cells along tumor-involved white matter structures using this method [40,45,46,47,48,49,50]. We further hypothesized that the presence of long tumor-associated tracts projecting outward from the tumor may be associated with poorer overall survival in glioblastoma. Ultimately, we identified specific patterns of tractography extending from tumors that appear to colocalize with microscopic and transcriptomic features of invasiveness. While the true relationship of observed tumor-associated tractography and underlying tumor biology remains unclear, the demonstration of patterned diffusion and identification of quantitative features associated with overall survival merits further work into the underlying biology of glioma invasion, clinical stratification, and development for possible local delivery of therapeutic agents.

## 2. Materials and Methods

This investigation was carried out in several steps including both human and animal model experiments. First, a prospective human cohort with newly diagnosed glioblastoma was recruited and assessed using a high-angular-resolution QSI pulse sequence in addition to standard-of-care imaging. Imaging from this cohort was qualitatively assessed for initial tractography pattern identification and refining methods. Then, a historical cohort of patients with glioblastoma who underwent scans with a similar pulse sequence was analyzed quantitatively to assess the prognostic importance of several tractography-derived metrics on patient survival. To further test our hypotheses, RNA sequencing analysis of tract-associated and non-tract-associated samples from an individual in the initial human cohort was carried out to assess for differences in transcriptomic expression between sample groups. Finally, to substantiate the claim that tractography features may colocalize with invading tumor cells, patient-derived xenograft mouse brains were scanned using an adapted QSI protocol, and 3D reconstruction of tumor cells in one of these was qualitatively compared with QSI-derived tractography. Methods for each of these experiments are explained as follows.

### 2.1. Image Acquisition Through QSI

Past work has developed and validated dwMRI methods designed to map biological architecture and relate these architectural templates to physiological function [3,40,44,49,50,51]. Through the application of QSI and associated methods [43], magnetic resonance signals are derived from proton diffusion and can be represented as a function of the strength and duration of the diffusion encoding magnetic field gradients. These variables may be probed during diffusion sampling to obtain a probability distribution function (PDF) for diffusional events, thereby demonstrating the angular resolution needed to discriminate the underlying fiber populations in whole tissue. The data points contained in each voxel are equal to specific q values corresponding to the image from which it was taken. The areas of PDF protrusion represent directions of maximum diffusion (i.e., maximum intravoxel coherence). Multi-voxel tract generation is carried out using a sub-voxel tri-linear streamlining algorithm under Eulerian integration, modified so that multiple fiber orientations may be discerned per voxel. Inter-voxel tracts may be selectively generated that intersect a region of interest through orientational coherence defined by a range of permitted angles. The degree of tract angle separation constitutes the underlying basis of the tract orientation formed through the coherent linkage of diffusional maxima in one voxel with the diffusional maxima of similar orientations in neighboring voxels. The outcome of this formulation is a geometrically patterned diffusional tractography rendering reflective of the underlying fiber pattern independent of the tissue source. This method has generally been previously applied to the derivation of tissue architecture in humans and mice [40,44,51].

### 2.2. Human Imaging Dataset Acquisition and Post-Processing

All patient participants in the primary enrollment cohort provided informed consent prospectively (IRB #1097531). Patients presenting with volume-occupying brain lesions on previous computed tomography (CT) to a single tertiary care center were approached for consent to undergo a research-specific dwMRI imaging protocol incorporating standard single-shot EPI spatial encoding on a siemens 3T scanner (b-value = 1000 s/mm^2^, 64 gradient directions, echo time = 95 ms, voxel size 1.8 × 1.8 × 4.0 mm), in combination with anatomical MRI. These scanning parameters were selected to maximize the likelihood of observing any structures with coherent diffusional property oriented nearby the tumor, while endeavoring to minimize the scan time given that all diffusion data were acquired as add-on research sequences during standard-of-care preoperative scans after the diagnosis of new brain tumors. Of successfully enrolled subjects (*n* = 13), WHO Grade 4 glioblastoma diagnosis was confirmed on postoperative pathology in *n* = 7 cases, confirming inclusion for the present study. Tumor region(s) of interest (ROI) were manually segmented within diffusion spectrum imaging studio (DSI Studio, Version Date: 3 October 2019) after registration of QSI and post-contrast T1- or T2-weighted images. Tumor ROI were generated by manually tracing a single boundary encompassing the enhancing tumor and necrotic core, but excluding any evident peritumoral edema, on each axial slice. In cases of multiple tumor foci, each was segmented and investigated individually. T1-weighted contrast-enhanced images or T2-weighted images were favored for aiding tumor segmentation when available. While explicit distortion correction of raw dwMRI data was not employed as part of pre-processing, the bad slice inspector tool was used when loading data into DSI Studio, prior to masking to filter out background, in an effort to identify and eliminate possible head motion or eddy current-related artifacts. “Bad slices” were identified and excluded from further processing in only a minority of cases.

The post-processing of QSI data was then completed in DSI studio for each scan to generate tumor-associated tractograms. After setting tracking parameters including a tract length threshold of 0–300 mm, angular threshold of ≥45°, and step size of 0.3 mm, diffusional tracts which explicitly intersected the manually segmented tumor ROI were displayed. Tumor-intersecting tractography or “tumor tracts” were observed qualitatively across the study cohort, and metrics quantified within DSI Studio included mean tract length and mean projecting tract length. The mean tract length was defined as the mean length of all post-processed tumor-intersecting tracts. The mean projecting tract length was defined as the mean length of those tracts which extended outward from the tumor ROI rather than terminating within the segmented mass or adhering closely to the surface of the mass, as described by Taylor et al. as “core” and “surface” tracts, respectively [40]. Such tracts were manually selected from the generated tractograms for further quantification. Thus, the mean projecting tract length specifically excluded any tracts which were confined exclusively within or circumferentially around the tumor ROI. Both tract length metrics were computed directly by DSI Studio. Tumor volumes were additionally quantified using 3D Slicer (v4.10.2). Patient demographics, receipt of surgery and/or chemoradiation, MGMT promotor methylation status, IDH mutation status, tumor location, number of tumor foci at presentation, and patient outcomes were obtained from the electronic medical record.

For the confirmation that observed tractography patterns were also seen more generally across a larger cohort, and for a more rigorous analysis of association with clinical outcomes, a de-identified historical comparison dataset from University of Texas, MD Anderson Cancer Center (MDACC) was utilized. Ethical approval was obtained for use of retrospectively collected, anonymized data (IRB #MDACC-RR–03–800). This dataset was selected because it contains a moderately sized cohort of patients who had undergone scans which employed comparable (though not identical) acquisition parameters in terms of b-value and number of gradient directions. Diffusion-weighted images were acquired using single-shot echo-planar (EPI) spatial encoding pulse sequence with voxel size of 0.86 × 0.86 × 3.5 mm, 27 gradient directions plus b-zero, and b-value of 1200 s/mm^2^. Diffusion-weighted and accompanying T1-weighted and/or T2-weighted sequences were acquired using 3.0T GE Signa HDxt MRI scanners with 8-channel high resolution brain coil (GE Healthcare, Waukesha, WI, USA). Of 79 patients treated for glioblastoma between 2008 and 2013, *n* = 66 comprised the final analysis cohort after excluding those with insufficient preoperative imaging for the present analyses, infratentorial tumor locations, and/or intraventricular tumor extension. The same image postprocessing methods as described above were employed, except that for cases of multifocal disease, only the largest was segmented and used to generate tumor-intersecting tractography. Additionally, the number of projecting tract bundles was also quantified as an additional metric, again using DSI Studio software. In addition to QSI-derived quantitative features associated with observed tractography patterns, intra-tumoral dwMRI metrics including quantitative anisotropy (QA), fractional anisotropy (FA), mean diffusivity (MD), and axial diffusivity (AD) were calculated within the segmented tumor ROI. For this retrospective anonymized dataset, molecular characterization of tumors including MGMT promotor methylation and IDH mutation status was not available, though age and basic treatment details are reported in a pooled fashion.

### 2.3. Peritumoral Tract Sampling and RNA Sequencing

One of seven patients enrolled at the primary enrollment center underwent additional study collection of tumor-associated samples intraoperatively for grouped analysis using RNA sequencing (RNAseq). Samples were chosen using preoperative MRI with overlaid tractography combined with stereotactic intraoperative navigation system (BrainLab, Munich, Germany). A total of 13 samples were collected and classified as either normal peritumoral brain tissue (*n* = 2), peritumoral tract-associated tissue (*n* = 6), or tumor tissue (*n* = 5). RNAseq was performed using Illumina HiSeq2500 (GeneWiz/Azenta Life Sciences, South Plainfield, NJ, USA). The 150-nucleotide sequence reads were aligned to the hg38 build of the human genome using gsnap [52]. The genomic locations of genes/exons defined in Refseq were extracted from the refGene.txt (http://hgdownload.cse.ucsc.edu/goldenPath/hg38/database/refGene.txt.gz, accessed on 15 January 2019). Read summarization at the gene level was performed with in-house scripts using reads with a mapping quality of 20 or greater. After discarding genes with fewer than 5 reads-per-million in at least 2 samples, we retained 15,562 genes for analysis. Clustering of samples based on expression profiles was performed with the hclust routine of R (Posit Software, Boston, MA, USA). This analysis showed that the samples acquired from the tumor areas that correspond to tracts cluster separately from the bulk tumor or control samples. Gene expression was analyzed in R with the limma package [53,54], applying voom precision weights to account for the mean-variance dependency observed in the standardized read counts [55]. *p*-values were adjusted with the Benjamini–Hochberg method to control the false discovery rate and account for multiple hypothesis testing [56]. A 2-fold difference was imposed as minimal threshold for differential expression. Functional classification was carried out using gene set enrichment analysis (GSEA) with the Gene Ontology annotations at http://pantherdb.org, accessed on 15 January 2019 [57].

### 2.4. Patient-Derived Xenograft (PDX) Models, Image Acquisition, and Microscopy

In this study, *n* = 5 immunocompromised 8-week-old Nu/J male mice (Jackson Laboratories, Bar Harbor, ME, USA) were injected sub-hippocampally with patient-derived GSCs using a stereotaxic apparatus (Kopf, Tujunga, CA, USA). Each mouse was initially sedated with 4% isoflurane and maintained with 2% isoflurane. After leveling the skull, a hole was drilled with a #72 micro drill bit (Kyocera, Kyoto, Japan) at coordinates −2.0 mm AP and +1.5 mm ML relative to Bregma. A 75 RN Hamilton syringe was then lowered to a depth of −2.5 mm DV at a rate of 0.5 mm per minute, and 500,000 primary GSCs resuspended in a total volume of 3 μL were injected at a rate of 0.5 μL per minute using a Stoelting Quintessential Stereotaxic Injector (Stoelting Co, Wood Dale, IL, USA). To reduce backflow, the syringe rested for 2 minutes post-injection, before withdrawal at a rate of 0.5 mm per minute, and the cavity was immediately sealed with bone wax (Ethicon Inc., Raritan, NJ, USA).

After allowing tumors to grow for two months, mice were sacrificed, and whole brains were removed and fixed in 10% phosphate-buffered formalin. Preserved brains were scanned with a modified QSI protocol employing a multi-shot EPI pulse sequence on a Bruker 7T (BioSpec, Billerica, MA, USA) scanner with a Cryoprobe, applying 512 gradient directions, a b-value of 1200 s/mm^2^, an echo time of 16.7 ms, a voxel size of 0.16 × 0.16 × 0.16 mm, and 8 k-space segments. Tumor-associated tractography was generated based on manually drawn ROI in DSI Studio. One of the imaged mouse brains was then sectioned and immuno-stained for human nuclear antigen to highlight human cell nuclei in the xenograft brain tissue, and images converted to a binary map of only the stained cells using Amira Software (v2020.2, Thermo Scientific, Waltham, MA, USA). Tumor cells were then manually segmented to remove background, and a map of segmented tumor cells reconstructed in 3D for qualitative comparison with tractography.

### 2.5. Statistical Analyses

Population characteristics are summarized using descriptive statistics. The association of continuous variables with overall survival was assessed using linear regression with a report of β coefficient and *p*-values, while binary variables were assessed by comparing the survival time means using two-sided *t*-tests after stratifying the population in terms of each variable. Applying these methods, age, tumor location, volume, calculated intra-tumoral dwMRI metrics, and QSI-derived tractography features were examined for possible outcome associations in the historical comparison cohort. Kaplan–Meier curve analysis was also utilized to visualize differences in survival trajectories with respect to age, tumor volume, MD, mean tract length, mean projecting tract length, and number of projecting tract bundles. All Kaplan–Meier curves are stratified by population quartiles in terms of each predictor variable, and survival distributions between quartiles were assessed using log-rank tests. Statistical analyses were carried out in Python (v3.6.5, Python Software Foundation, Wilmington, DE, USA) and R/RStudio (2024.04.2 + 764, Posit Software, Boston, MA, USA).

## 3. Results

### 3.1. Primary Enrollment Cohort

A total of seven patients with pathology-confirmed glioblastoma with a mean age of 66.9 (±10.6) years, 57.1% male, were included in the pilot study of 64-gradient-direction QSI. Tumor locations were predominantly left-sided (5/7, 71.4%) with most located in the temporal or parietal lobes. Three patients (42.9%) had multifocal disease at presentation. Other population characteristics can be found in Table 1.

Tumor-associated tractography generated a consistent pattern of projecting tracts oriented outwardly from the tumor surface (defined in diffusion imaging by core–shell phenomenon, as described previously by Taylor et al.) [40]. Tumor ROI varied in size according to tumor volume, with a mean volume of 18.7 (±4.2) cm^3^. The associated mean overall tract length was 23.2 (±3.1) mm, while the mean projecting tract length was 38.9 (±4.9) mm. The workflow for generating tumor-associated tractography as well as several representative examples of the observed pattern are depicted in Figure 1.

In instances of multifocal tumor present on presentation (*n* = 3), projecting tractography was anecdotally observed to span between foci of tumor ROI (Figure 2). Given that longitudinal, diffusionally coherent tractography bundles in the brain usually correspond to white matter structures, this observation raises the possibility of functional connectivity between regions of distinct tumor foci in instances where patients present with multifocal disease. Taken together, these findings show that, using QSI, we can define tumor-associated tracts extending from tumors and, in the case of multifocal tumors, observe possible functional connectivity between distinct foci.

### 3.2. Historical Comparison Cohort and Analysis of Outcomes

Sixty-six patients with pathology-confirmed glioblastoma with a mean age of 59.4 (±12.1) years were included from the historical comparison cohort at MDACC. Compared to the primary enrollment cohort (*n* = 7), a larger proportion of cases had right-sided (33/66, 50%) or non-cortical (7/66, 10.6%) tumors, and the mean tumor volume was 43.1 (±37.9) cm^3^ (Table 2). Overall, a qualitative observation confirmed that the same pattern of projecting tractography that was observed in the smaller pilot cohort was also seen more generally in the larger cohort. The mean tract length was 29.1 (±11.7) mm, and the mean projecting tract length was 64.2 (±19.6). On average, each tractography demonstrated 5 (±2.1) distinct tract bundles projecting outward from the tumor ROI, with some of these corresponding to specific white matter pathways of known anatomic and functional importance (as shown in Table 2).

Univariate linear regression analyses demonstrated statistically significant positive associations of MD and AD intra-tumoral dwMRI metrics with longer overall survival measured in days (β = 504.75 [28.20, 981.29], *p* = 0.038; β = 474.17 [30.69, 917.65], *p* = 0.037). Stratifying the population according to several binary categorical variables, significant negative associations with survival were noted for tumors involving the corpus callosum (*p* < 0.001), left-sided tumors (*p* = 0.029), and tumors involving the inferior longitudinal fasciculus and/or inferior frontal-occipital fasciculus (*p* = 0.008). While significant associations between overall survival and tractography-related metrics including mean tract length, mean projecting tract length, and number of projecting tract bundles were not observed on the linear regression analysis, Kaplan–Meier survival statistics comparing the cohort stratified by quartile demonstrated significant negative effects on the overall survival of older age (*p* = 0.013), longer mean total tract length (*p* = 0.039), and longer mean projecting tract length (*p* = 0.022). Full Kaplan–Meier analyses including curves for these comparisons as well as stratification by tumor volume, MD, and number of projecting tract bundles observed are depicted in Figure 3.

### 3.3. Transcriptomic Analysis of Tract-Associated Tissue

After interrogating the utility of QSI as a prognostic tool, we investigated transcriptional changes between tract-associated brain tissue samples, tumor bulk, and non-tract-associated brain tissue to identify whether there are significant biological differences in cell populations associated with specimens defined by QSI. RNAseq demonstrated the overexpression of 528 transcripts in tract-associated brain tissue samples, versus tumor bulk and non-tract-associated brain tissue samples (Figure 4). A functional classification analysis of the 528 overexpressed transcripts was performed using GSEA, which revealed a significant overrepresentation of transcripts that regulate cell motility. A further analysis of the motility-associated transcripts revealed that they form significant functional regulatory networks, which implies an interconnected transcriptomic signature of cell invasion in cells that occupy the QSI-identified tracts. These results underscore the possible association of migratory GSCs with tractography bundles observed to be “projecting” outward from the tumor when tumor tractography is visualized.

### 3.4. Reconstruction of GSC Distribution in PDX Mice Mirrors Tumor-Associated Tractography

Application of QSI methods in PDX mice (*n* = 5) produced a similarly conserved pattern of tumor-associated tractography coursing over and under, but not through, the hippocampus beneath which primary human GSCs were injected during PDX model implantation. Aside from the qualitative conservation of the tractography pattern between imaged mice, the segmentation and three-dimensional reconstruction of tumor cells within the PDX brain of *n* = 1 mouse additionally demonstrated a strikingly similar spatial pattern, including apparent crossing of the midline via the corpus callosum (Figure 5).

## 4. Discussion

We provide preliminary evidence that QSI-generated tractography may harbor quantifiable features which correspond to localized cellular invasion in glioblastoma. While only performed in one individual, RNAseq results comparing tract-associated samples to bulk tumor and tract-unassociated samples highlighted the significant over-expression of cell motility-related transcripts, whereas a clustering analysis correctly separated samples according to distinct transcriptomic profiles between groups (Figure 4). Furthermore, in an individual PDX mouse, immunostained GSCs, when spatially reconstructed, mirrored the pattern observed in the tumor-associated tractograms previously captured from the same specimen (Figure 5). The observed tractography patterns were replicated across all individuals in the study, spanning both PDX mice (*n* = 5) and patients from two institutions (*n* = 73 total). In the larger historical cohort, which employed a similar QSI acquisition technique to the primary enrollment cohort, mean tract length and mean projecting tract length from individual tumors predicted overall survival after diagnosis across the population when stratified in quartiles, with longer observed tracts portending shorter overall survival (Figure 3).

Extending prior work demonstrating the tendency of glioblastoma cells to invade and mediate tumor recurrence both local to the resection cavity and at distal sites across the brain [6,7,8,9], this study reinforces the concept that dwMRI techniques, including QSI and high-angular-resolution techniques, as well as potentially more clinically feasible techniques such as DTI, may have the capacity to detect tumor-specific migration pathways. Diffusion imaging techniques have been developed that reliably map intracerebral white matter pathways [32,33,34,35]. While we do not yet know the degree to which QSI (and other dwMRI techniques) map alterations in the architecture of white matter induced by migrating tumor cells, our method of post-processing imaging using a segmented tumor as the ROI for tract generation confirms that the pathways observed, quantified, and associated with patient outcome in this study are in fact tumor-associated. Indeed, our method in both mice and humans was designed to capture longitudinally oriented diffusivity patterns within the tissue which traverse, intersect, or about the segmentation-defined tumor ROI. This substantiates the framework that the observed phenomena are associated in some manner with the underlying tumor biology.

### 4.1. Clinical Prognosis Based on Tractography-Derived Metrics

In the historical cohort from MDACC, we individually observed and documented the regional overlap of observed tumor-specific tractograms with known white matter bundles of anatomic significance (Table 2). From an analysis of patient outcomes, tumors which occupied the corpus callosum had significantly worse outcomes than those who did not (*p* < 0.001), while those who had tumors located elsewhere but with tractography crossing over to the opposite hemisphere also demonstrated a trend towards worse outcomes (*p* = 0.09). This is consistent with the prior literature noting corpus callosum involvement as a poor prognostic indicator in glioblastoma [58,59,60]. Similarly, patients whose tractography overlapped with the inferior longitudinal fasciculus and/or inferior frontal-occipital fasciculus had a worse prognosis (*p* = 0.008). These results lend support to the idea that the specific tract impacted by the tumor, depending on its location of origin, may matter for patient outcomes, as has been reported previously [39]. This may be due to location-related functional impairments, or perhaps because cell invasiveness is more efficient in or along prominent axon bundles due to interactivity with resident normal cells [19,20]. Furthermore, metrics including a longer mean tract length and longer projecting tract length were also associated with shorter overall survival. More extensive tumor cell invasion, or greater opportunity for wide-spread invasion through the presence of more (or longer) white matter bundles intersecting the tumor, may be associated with earlier progression of disease. While progression free survival was not separately tracked as an outcome in this study, earlier progression may itself lead to shorter overall survival, explaining the observed results. Future investigation of disease progression and recurrence patterns relative to pre-operative diffusion tractography would provide additional information useful for understanding this relationship.

Other investigations of post-processed tractograms generated from dwMRI sequences have recently been published, lending support to the notion that tumor-associated tractography may be associated with tumor cell migration [39,40,42]. In particular, Rammohan et al. presented an FA-derived composite white-matter score which showed prognostic significance in MGMT-unmethylated glioblastoma [39] and Taylor et al. examined the core–shell ratio (length of tracts within the “necrotic core” of the tumor compared to the length of the tracts which circumscribe the mass at its visible margin) [40]. Of note, Taylor et al. carried out their study using the same MDACC cohort combined with a small number of cases available through the Cancer Genome Atlas, and found that a higher core–shell ratio was correlated with poorer overall survival. Motivated by the hypothesis that more extensive tumor-associated tractograms may be associated with the long-distance migration of glioblastoma cells, the present study focused on the quantification of features which account for tracts reaching beyond the tumor margin inferred from routine neuroimaging in addition to conventional intra-tumoral dwMRI metrics (e.g., MD, FA) [37,61,62]. While mean (total) tract length captures all core, shell, and projecting tracts generated through the tracking algorithm, the mean projecting tract length captures only tracts manually selected as projecting away from the “shell” of the tumor upon the analysis of processed tractograms. Longer tract length, in terms of both metrics, was significantly associated with poorer overall survival (Figure 3). Additional meaningful prognostic metrics could also perhaps be uncovered if diffusion parameters were systematically varied to optimize ability to distinguish between tissue structures expressed at varying spatial scales [63,64,65,66]. Such variations could include evaluating algorithms where multiple b-values (rather than one b-value) could be employed, either varying the b_max_ or employing multiple shells of diffusion acquisition. Alternately, varying spatial resolutions (e.g., voxel sizes) and number of gradient directions could be used to further minimize signal-to-noise in acquired images. In the present study, these parameters were fixed at specific values as part of pre-planned pulse sequences such that individual parameters were not subject to investigation. Such future directions may hold benefits for dwMRI and tractography analyses of heterogeneous tumors like glioblastoma in which different tissue components (e.g., tumor core versus shell, necrotic regions, tumor margin, and surrounding parenchyma containing invading cells) have varied diffusional properties corresponding to differing extents of migration and cellularity [6,16,17,18].

On the other hand, higher intra-tumoral mean and axial diffusivity values were associated with longer overall survival, which has been reported previously [62]. While the interpretation of these metrics in the context of a variably necrotic core in high-grade glioma may be difficult to discern, less intact architectural boundaries and greater “liquification” within the assessed tumor ROI may lead to reduced directional diffusivity (perpendicular or axial to involved fiber bundles within the tumor), and therefore portend greater migration activity away from the tumor core and poorer outcomes [67]. This explanation would explain both findings in the current study (i.e., lower mean and axial diffusivity in addition to longer total and projecting mean tract lengths) with respect to our tract-associated cellular migration hypothesis. Even so, studies have suggested that mean diffusivity does not always correlate uniformly with tumors that typically have more intra-tumoral architectural integrity and/or less necrosis; conventional dwMRI metrics alone may perform better at predicting outcomes (or pathology) when combined with other radiomic markers [36,38,68]. Further study of tractography-derived versus basic dwMRI metrics head-to-head would be an interesting future direction as new tractography-derived analyses emerge. Additionally, the presence of edema around the tumor could theoretically introduce a free water effect which affects diffusion metrics particularly adjacent to invasion-affected white matter tracts, which are hypothesized to coincide with observed tumor-intersecting tractography. Accordingly, any observed directionality may be mitigated because of that edema, reducing directional specificity. Algorithms which incorporate free water elimination or multi-tissue techniques could help adjust for any possible signal loss due to invasion-associated edema.

Finally, while we observed statistically significant associations of tractography-derived metrics with survival time, the implications of the location and distribution of invading tumor cells within the brain on glioblastoma outcomes remain preliminary. Further, it is possible, given all patients in this study had invasive disease by pathologic definition, that certain outcome associations may be non-significant specifically because of homogenous disease severity, or that the observed significant correlations of outcomes with certain tract pathways (corpus callosum, inferior longitudinal fasciculus, and/or inferior frontal-occipital fasciculus) and overall tract length are actually driven by the likelihood of earlier recurrence or functional deterioration when those pathways are involved. Future clinical outcome studies should perform multivariable analyses which adequately control for multiple additional variables including recurrence time, recurrence location, symptomatic progression, chemoradiation treatment effects, and others which were not collected by the present study, and which may better explain the relationships observed within the present univariate analyses.

### 4.2. Tracking Tumor Cell Migration: Existing Evidence and Clinical Implications

In addition to studying clinical outcomes (i.e., survival) and its correlation with novel tractography-derived metrics, we attempted to explore the potential pathologic underpinnings of tumor-associated tractography with respect to migrating cells through RNAseq and PDX model analyses (Figure 4 and Figure 5). Previous work visualizing migrating tumor cells in vivo shows some of the same migratory patterns as our results on tumor-associated tractography and microscopy in this study (including the hippocampus being a cancer cell-free zone in the PDX model), further supporting the validity of the presently observed association between tumor tractography and migrating cells [69,70]. Our results collectively support the further study of the possibility that migratory glioblastoma cells may co-localized with observable tumor-intersecting tracts. This future work is important because tracking glioblastoma cell migration more precisely in a non-invasive manner may open the door to targeted therapeutics in the future [29]. For example, confirmation that specific white matter bundles are riddled with migrating cells, which may mediate recurrence, could enable future therapeutic strategies selectively targeting these tracts. Possible localized therapeutic modalities could include radiotherapy, convection enhanced delivery, laser interstitial thermal therapy, and others still in development which may not yet have been introduced clinically [71,72,73,74]. Although this is an exciting possibility for the future, significant additional work is needed, and currently tractography is not a consideration for routine radiotherapy treatment planning [71,75,76,77]. While the clinical application of diffusion tractography currently centers around identification of functionally important white matter pathways near to the tumor which may be identified preoperatively and spared during maximal safe resection [32,33,34,35], targeting tumor-infiltrated tracts would generate new treatment approaches in this disease. The extent to which this technique can be used for live migration monitoring is uncertain at best, given that there have been no longitudinal studies examining multi-timepoint QSI to assess tumor-associated tractography. Yet, the finding that individual tract bundles may sometimes span multifocal ROI within a single patient (Figure 2) provides anecdotal evidence which encourages the follow-up of this hypothesis. Furthermore, emerging evidence suggests that glioma cells play an active role in remodeling the extracellular matrix to promote localized invasion through brain tissue and eventually mediate disease recurrence [78]. The extent to which diffusion-derived tractography is biologically related to existing white matter bundles and/or invasion pathways involving ECM remodeling remains unclear. The latter of these, if differentiated from the first, may lend an aspect of disease specificity to the observed tractographic features, which could aid treatment targeting, if confirmed.

### 4.3. Limitations

This study is subject to several important limitations. First, the small number of individual cases used in the RNAseq (*n* = 13 samples from an individual patient) and PDX mice (*n* = 5) limit the generalizability and conclusiveness of these findings—these aspects in particular must be confirmed in follow-up reports applying similar methods. The GSEA presented in this study utilized samples from tract-associated samples related to the tumor versus non-tract-associated samples from within and nearby the tumor, but we did not directly sample from white matter tractography unrelated to the tumor, which would have been an additionally helpful negative control for interpreting our findings more conclusively. While challenging to accomplish in humans due to lacking clinical justification from sampling healthy tissue distant from the tumor, such studies could be performed in animal models; precise regional biopsy on such a small scale is challenging and was not immediately possible in the context of the current project. Further, the software used for analysis, DSI Studio, permits a range of tracking algorithm parameter selections for generating tractograms; those used here (angular threshold of 45° and maximum tract length of 300 mm for the historical cohort used in outcome analyses) were particularly permissive. Tumor ROI segmentation is also subject to possible bias especially when carried out manually based on a single co-registered MR sequence, given known limitations to distinguishing the tumor boundary on any single sequence; additional multi-modal segmentation tools have since emerged which could also be employed in future studies through more accurate segmentation [79,80,81]. While survival analyses yielded statistically significant results when conducted on *n* = 66 patients from a single center, the reproducibility of our findings should be evaluated with larger, multicenter cohorts. We included data from two institutions in the present study; however, the data presented from the primary enrollment cohort (*n* = 7) should be considered as a pilot dataset; the MDA dataset serves as a larger dataset that qualitatively validates the observations of the smaller cohort, but must still be quantitatively compared with other similarly datasets to be generated in the future. The two cohorts could not be quantitatively compared in the present study owing to the small size of the principal enrollment cohort, limiting the ability of the current study to evaluate accuracy and within-group variability of results across different settings. A method which could be applied in future work to evaluate between-group variability in mouse or human imaging studies has been previously presented by Giaccone et al. [82]. Methodologically, future studies should also investigate the effect of differences in post-processing software, tumor segmentation algorithms, image acquisition, and the timing of image acquisition. Further studies should also compare different dwMRI parameters while employing similar post-processing methods. Modifying the b-value, number of gradient directions, and other parameters could enable the optimization of the scan time required to achieve the same results. Finally, longitudinal studies employing serial QSI examinations to assess tumor recurrence over time would be an interesting follow-up study.

## 5. Conclusions

QSI may discriminate tumor-specific patterns of inter-voxel coherence that may represent white matter pathways susceptible to tumor cell invasion, consistent with other recent work showing the prognostic value of features derived from tumor-associated diffusion tractography. The findings of this study contribute to the groundwork for future studies on therapeutic targeting and patient stratification based on alterations in extra-tumoral tissue architecture in glioblastoma. Future studies applying the same imaging and post-processing techniques in human and mice must be conducted to quantitively evaluate the reproducibility and variability of our presented observations across different practice settings and in larger populations.

## Figures and Tables

**Figure 1 cancers-16-03669-f001:**
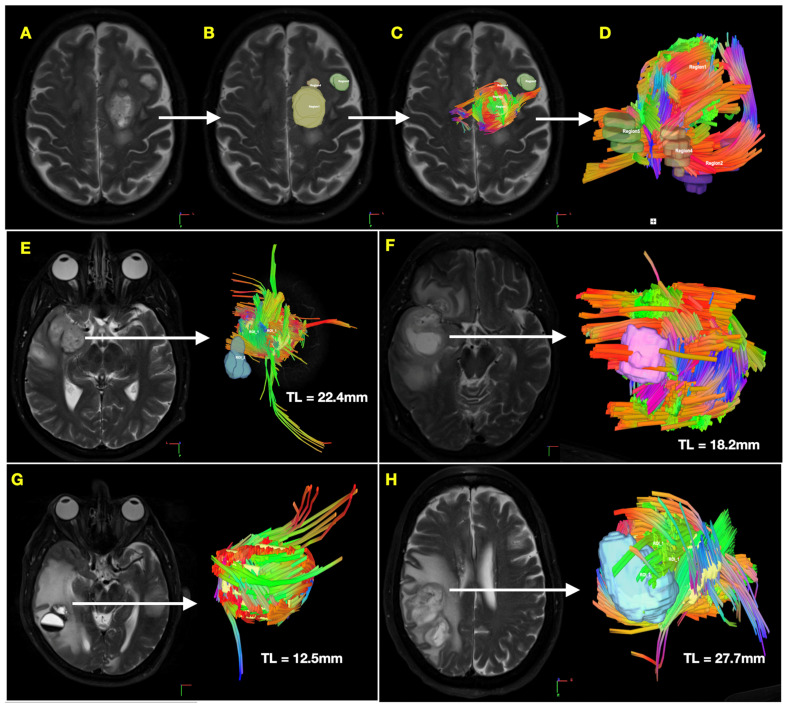
Tumor-associated tractography patterns in glioblastoma identified using QSI. Demonstration of tumor segmentation and visualization of tumor-associated projecting tractography derived from generalized q-space imaging (QSI, 3T, 64 gradient directions, b = 1000 s/mm^2^). Panels (**A**–**D**) demonstrate the workflow for manually segmenting the tumor regions of interest (ROI) using T2-weighted MRI registered with QSI (**B**). Colors represent the orientation of the vector: blue denotes superior–inferior directionality, red denotes left–right directionality, and green denotes anterior–posterior directionality. Three-dimensional tumor-intersecting tractography is displayed using pre-set tracking parameters (angular threshold of 45°, step size of 0.3 mm, maximum tract length of 300 mm, (**C**–**D**)). Four additional representative examples of tumor-associated tractography are depicted in (**E**–**H**), with T2-weighted MRI axial image demonstrating the tumor shown to the left of the arrow and the post-processed three-dimensional tumor ROI with intersecting tractograms shown to the right of the arrow within in each pane. Mean tract length (TL) is reported for each.

**Figure 2 cancers-16-03669-f002:**
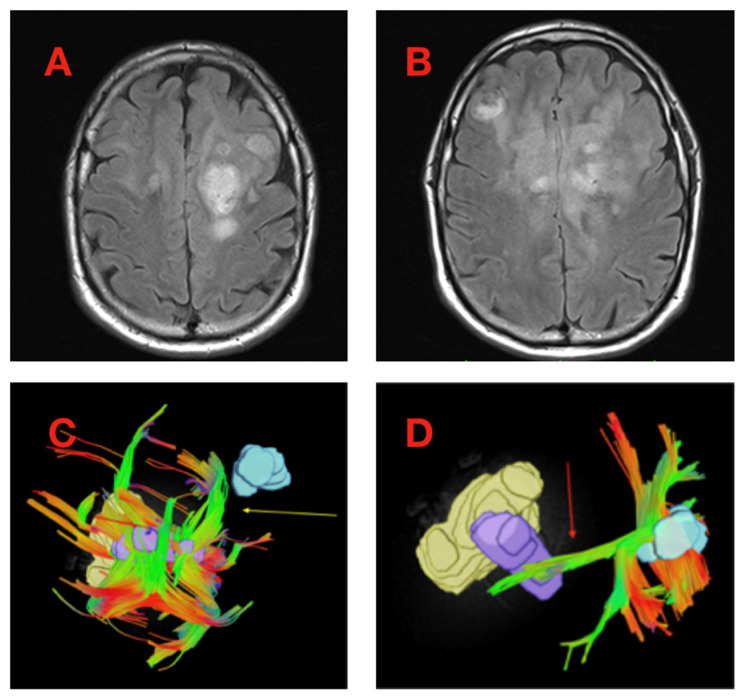
Projecting tumor-associated tractography observed bridging between distinct tumor foci in multifocal glioblastoma. In a minority of cases presenting with multifocal glioblastoma, tumor-associated tractography sometimes demonstrated directionally oriented fiber bundles passing between anatomically distinct tumor regions of interest suggesting possible functional connectivity between the regions. An example of this phenomenon is illustrated from a single patient from the Rhode Island Hospital cohort. FLAIR images depict multifocal disease at two axial levels, (**A**,**B**). After generating tumor regions of interest (ROI) and associated ROI-intersecting tractography, extending tract bundles appear to span between distinct tumor foci (arrows in (**C**,**D**)). In this case, two separate tumor ROI were separately evaluated with “reciprocal” tracts observed to span between the two ROI.

**Figure 3 cancers-16-03669-f003:**
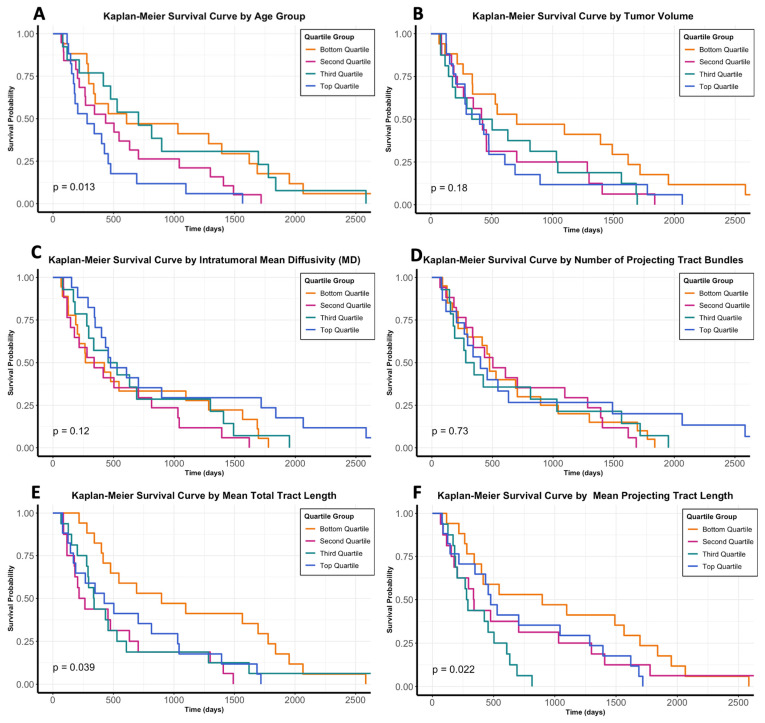
Survival analysis of tumor-associated quantitative imaging features. On Kaplan-Meier survival analysis stratifying the historical glioblastoma cohort from MDACC (*n* = 66) by quartile groups in terms of select demographic and radiomic metrics (*n* = 66): age (**A**), tumor volume (**B**), intratumoral mean diffusivity (**C**), number of projecting tract bundles observed (**D**), mean total tract length (**E**), and mean projecting tract length (**F**). Statistically significant negative effects on overall survival were seen with older age (*p* = 0.013), longer mean total tract length (*p* = 0.039), and longer mean projecting tract length (*p* = 0.022). Larger tumor volume and lower mean diffusivity were non-significant but trended towards negative effects on survival as well.

**Figure 4 cancers-16-03669-f004:**
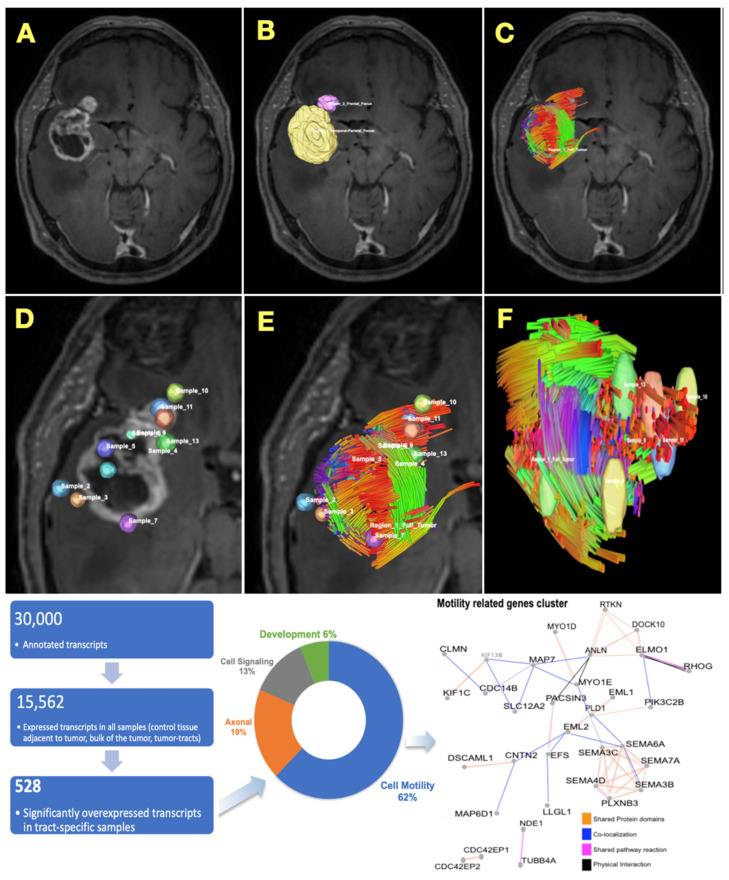
Transcriptomic analysis of tractography-associated peritumoral tissue biopsies. T1-weighted contrast-enhanced MRI image of right temporal glioblastoma (**A**) was overlaid with 64-direction QSI to generate tumor-associated tractography (**B**,**C**). Regions of interest associated with peritumoral tissue with and without associated tracts, as well as within the bulk of the tumor, were then identified and segmented (**D**–**F**). Intraoperative stereotactic biopsies of tissue at these 13 locations were collected during resection, placed immediately in liquid nitrogen, transferred to RNAlater-ICE frozen tissue transition solution for RNA integrity maintenance. Full RNA sequencing was then performed on isolated total RNA extracted from samples, and 528 transcripts which were significantly over-expressed (≥2-fold) in tract-associated samples (versus tumor bulk or peritumoral non-tract-associated samples) were identified. Functional clustering of these genes showed a significant representation of cell motility-related transcripts (62%, bottom panel), while regulatory network analysis revealed a transcriptomic network that modulates cell motility within the cells that occupy the QSI-identified tracts in a patient with glioblastoma.

**Figure 5 cancers-16-03669-f005:**
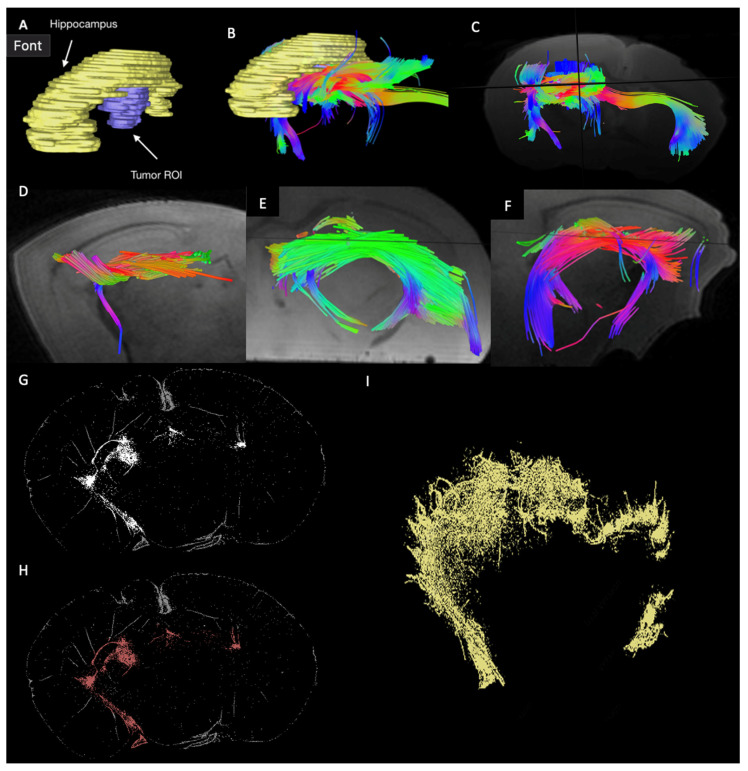
Three-dimensional reconstruction of glioblastoma stem cell (GSC) distribution in patient-derived xenograft mouse model recapitulates tumor-associated tractography phenotypic pattern. *n* = 5 immunocompromised Nu/J mice were injected sub-hippocampally with patient-derived GSCs, sacrificed after two months, with whole brains resected and fixed, and preserved brains scanned using QSI (Brüker 7T Scanner with cryoprobe, 512 gradient directions) to generate tumor-associated tractography. While (**A**–**C**) demonstrate the process of segmenting the hippocampus and ROI and displaying associated tractography, (**C**–**F**) demonstrate the tractograms generated from four individual mice. Then, sectioned xenograft tissue was immuno-stained for human nuclear antigen to highlight human cells in the mouse brain, and images converted to a binary map of stained cells (**G**), manually segmented to remove background (**H**), and a map of segmented tumor cells reconstructed in three dimensions (**I**). The segmented mouse brain shown in (**G**–**I**) is the same as the imaged mouse shown in (**A**–**C**). Qualitative pattern conservation can be observed between imaging modalities (GSC spatial reconstruction with microscopy and QSI-generated tractography). In either case, tumor-associated tractography, or cells, are observed to course over, under, and around, but not through, the hippocampus.

**Table 1 cancers-16-03669-t001:** Rhode Island Hospital primary glioblastoma cohort patient characteristics. *n* = 7 total patients with newly diagnosed glioblastoma who underwent generalized Q-space imaging (3T, 64 gradient directions, b = 1000 s/mm^2^) between 2018 and 2019.

Patient Characteristics (*n* = 7)	*n* (%)/Mean (±SD)
Demographic Details	
Age (Years)	66.9 (±10.6)
Gender (Male)	4 (57.1%)
Treatment Details	
Surgical Resection (vs. biopsy)	6 (85.7%)
Chemotherapy (Temozolomide)	7 (100%)
Radiation	7 (100%)
Pathology Details	
MGMT Promotor Methylation	4 (57.1%)
IDH Wildtype	7 (100%)
Clinical Outcomes	
Mortality at Last Follow-up	7 (100%)
Progression-Free (Days)	141.0 (±94)
Overall Survival (Days)	308.5 (±182.3)
Tumor Location and Structural Features	
Location, Frontal	1 (14.3%)
Location, Temporal	3 (42.8%)
Location, Parietal	3 (42.8%)
Location, Occipital	0
Location, Corpus Callosum	0
Location, Basal Ganglia/Thalamus	0
Laterality, Left	5 (71.4%)
Tumor Volume (cm^3^)	18.7 (±4.2)
Number of Tumor Foci at Diagnosis	2.1 (±2.2)
Multifocal at Diagnosis	3 (42.9%)
Generalized Q-Space Imaging (QSI)-Derived Tractography Features	
Mean Tract Length (mm)	23.2 (±3.1)
Mean Projecting Tract Length (mm)	38.9 (±4.9)

**Table 2 cancers-16-03669-t002:** Associations of patient characteristics and imaging features with overall survival in University of Texas, M.D Anderson Cancer Center (MDACC) glioblastoma comparison cohort (*n* = 66).

Patient Characteristics (*n* = 66)	*n* (%)/Mean (±SD)	β (95% CI)	*p*-Value
Overall Survival (Days)	714 (±664.8)	–	–
Surgical Resection (vs. biopsy)	60 (90.9%)	–	–
Chemotherapy (Temozolomide)	63 (95.5%)	–	–
Radiation	64 (97.0%)	–	–
Multifocal Tumor at Diagnosis	10 (15.2%)	–	–
Demographics			
Age (Years, Mean ± SD)	59.4 (±12.1)	−13.24 (−26.55, 0.08)	0.051
Tumor Location and Structural Features			
Location, Frontal	22 (33.3%)	–	0.417
Location, Temporal	27 (40.9%)	–	0.853
Location, Parietal	29 (43.9%)	–	0.679
Location, Occipital	10 (15.2%)	–	0.496
Location, Corpus Callosum	4 (6.1%)	–	**<0.001**
Location, Basal Ganglia/Thalamus	3 (4.5%)	–	0.922
Laterality, Left	33 (50%)	–	**0.029**
Number of Tumor Foci	1.2 (±0.4)	61.37 (−338.88, 461.62)	0.760
Tumor Volume (cm^3^)	43.1 (±37.9)	−3.59 (−7.87, 0.69)	0.099
Calculated Intra-tumoral dwMRI Metrics (Mean ± SD)			
Quantitative Anisotropy (QA)	1.19 (±0.42)	139.85 (−253.72, 533.41)	0.480
Fractional Anisotropy (FA)	0.19 (±0.06)	−663.42 (−3433.88, 2107.04)	0.634
Mean Diffusivity (MD)	1.25 (±0.34)	504.75 (28.20, 981.29)	**0.038**
Axial Diffusivity (AD)	1.47 (±0.36)	474.17 (30.69, 917.65)	**0.037**
Generalized Q-Space Imaging (QSI)-Derived Tractography Features			
Identified Anatomic White Matter Structure Involvement (%)	51 (77.3%)	–	0.965
Corpus Callosum	34 (51.5%)	–	0.091
Corticothalamic Tracts	6 (9.1%)	–	0.461
Corticospinal Tracts	9 (13.6%)	–	0.691
Superior Longitudinal Fasciculus	19 (28.8%)	–	0.824
Inferior Longitudinal Fasciculus/Inferior Frontal-Occipital Fasciculus	20 (30.3%)	–	**0.008**
Uncinate Fasciculus	4 (6.1%)	–	0.589
Mean Tract Length (mm)	29.1 (±11.7)	−7.79 (−21.88, 6.31)	0.274
Mean Projecting Tract Length (mm)	64.2 (±19.6)	−5.63 (−13.98, 2.72)	0.183
Number of Projecting Tract Bundles	5.0 (±2.1)	12.21 (−47.47, 71.88)	0.684

Age, tumor location, volume, calculated intra-tumoral diffusion-weighted MRI (dwMRI) metrics, and generalized q-space imaging (QSI)-derived tractography features are reported in terms of proportions or means and standard deviations. Association of continuous variables with overall survival was assessed using linear regression with report of β coefficient and *p*-values, while binary variables were assessed using two-sided *t*-tests with reported *p*-values. The bolded numbers indicate *p* < 0.05 which is below significance threshold.

## Data Availability

Deidentified data used in this study may be available upon request to the corresponding authors.

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
