# Peer review of "Tumor-Associated Tractography Derived from High-Angular-Resolution Q-Space MRI May Predict Patterns of Cellular Invasion in Glioblastoma"

_cancers, 2024, doi:10.3390/cancers16213669_

Round 1

Reviewer 1 Report

Comments and Suggestions for Authors

The proposed study investigates the association between tractography in GBM patients and patterns of cellular invasion. I think the topic is very interesting and has important potential implications for GBM treatment. However, I think a few points should be addressed before publication.

My main concern is the use of QSI technique, especially given the relatively low b-value used in all experiments and the relatively limited number of diffusion gradient directions in the historical cohort. The choice of this method should be justified and more methodological details about post-processing and tractography should be given. Furthermore, if the authors want to claim that QSI and/or other HARDI techniques are necessary to predict migration pathways (see lines 417-19 in the discussion) they should compare results with simpler but more clinically feasible methods such as DTI.

Other minor points:

1) Abstract, line 31: give the number of patients in the first group

2) Introduction, lines 87-88: I think this description of dwMRI is too focused on tractography applications. The authors should describe in more general terms how dwMRI is sensitive to water molecule displacements. Also, diffusion restriction is not necessary to identify orientations

3) Methods, line 129: I think tractography "image" is not the most appropriate term

4) Methods, lines 137-139: please give more details about the acquisition protocol, e.g. spatial resolution, TE, etc.

5) Methods, line 147: why wasn't distortion correction performed? Please discuss

6) Methods, lines 148-150: this information was already given a few lines above

7) Methods, line 150: what do you mean by "diffusion rendering"? Also, give more details about the post-processing of QSI data and what information was used for tractography

8) Methods, line 225: scarified->sacrificed?

9) Methods, lines 227-229: please give more details about the acquisition protocol, e.g. spatial resolution, TE, etc.

10) Discussion, lines 462-466: please be more specific

11) Discussion, limitations: could the presence of tumour infiltration and increased free water affect tractography results? Could free water elimination or multi-compartment/multi-tissue techniques help improve tractography sensitivity and/or specificity in tumour areas? Please discuss

Comments on the Quality of English Language

The quality of English language is generally good and it doesn't impair understanding

Reviewer 2 Report

Comments and Suggestions for Authors

Dear Authors,

your manuscript talks about an interesting topic, which can offer important scientific insights with potential implications for clinical practice.

I ask you to answer the following questions:

- When selecting patients, were only non-mutated IDH patients considered as glioblastomas (as per the latest WHO classification)?

- In the materials and methods you talk about glioblastoma grade IV, while the new classification considers Arabic numerals (4)

- In the results you also considered how peritumoral invasion can influence progression free survival, can you better describe and argue this result in the discussion?

For the rest, the paper is well written, the images are clear, the conclusions are pertinent and the bibliography is well-corresponding.

Reviewer 3 Report

Comments and Suggestions for Authors

The title “Tumor‐associated tractography derived from high angular resolution Q‐Space MRI may predict patterns of cellular invasion in glioblastoma” is appropriate for this manuscript.

This article discusses the ability of QSI to discriminate tumor-specific patterns of inter-voxel coherence believed to represent white matter pathways susceptible to glioblastoma invasion. Future research on therapeutic targeting, patient stratification, and prognosis in glioblastoma may be enabled by these findings.

Overall, the paper is acceptable, but it should be improved and justified the methodology used by including more state of the art evidence as well as other small changes in the processing of the performance results of the proposed method.

More detailed comments are below.

The authors are strongly advised to increase the processing of the data analysis and therefore the performance of the results, following what was done in the case of [10.1007/978-3-031-13321-3_31] where the authors determined whether there were statistically significant differences between the means of two groups or more categorical groups, through a comparison of variability between groups. By determining which group is different from the others, it is possible to classify and therefore stratify the cases and to add validity from the performance point of view. It is important to note that in preliminary studies such as the one proposed and the one mentioned, it is highly important to provide as much proof as possible of the accuracy of a method, especially when the number of cases is as small as the cases in this study.

Also, it is recommended that the authors provide a recent update on how glioma actively regulates glioma extracellular matrix to create a microenvironment conducive to survival, invasion, progression, and resistance to therapy [10.1186/s12885-024-12751-3].

In the Materials and Methods section, the workflow of the study is unclear as both patients and mouse models are considered. The authors are requested to provide a clear order, and furthermore, a description of the sections should be anticipated before them in order to guide the reader towards a workflow as understandable as possible, avoiding confusion.

To make reading the document easier for the reader, an Abbreviations section should be added to the end of the document.

In conclusion, to make the article more coherent, the authors should expand the references and the English is fluent throughout the document.

Round 2

Reviewer 1 Report

Comments and Suggestions for Authors

The authors have addressed all the issues I raised in the previous review. The text with marked edits doesn't show any deletions text, even though it's obvious that the additions were intended as replacement of some of the previous text. Please make sure that the final text reads as intended.

Reviewer 3 Report

Comments and Suggestions for Authors

The title “Tumor-associated tractography derived from high angular resolution Q-Space MRI may predict patterns of cellular invasion in glioblastoma” is appropriate for this manuscript.
This article discusses the ability of QSI to discriminate tumor- specific patterns of inter- voxel coherence believed to represent white matter pathways susceptible to glioblastoma invasion. Future research on therapeutic targeting, patient stratification, and prognosis in glioblastoma may be enabled by these findings.

The authors have answered almost all comments in a comprehensive manner.
However, there is still one unresolved point.

Details below.

The increase in data analysis processing and therefore the performance of the results, following what was done in the case of [10.1007/978-3-031-13321-3_31], in whose work it was determined whether there were statistically significant differences between the means of two groups or more categorical groups, through a comparison of the variability between groups had been strongly commented and suggested to the authors. By determining which group is different from the others, it is possible to classify and therefore stratify the cases and add validity from the point of view of performance.  It is important to note that in preliminary studies such as the one proposed and the one mentioned, it is extremely important to provide as much evidence as possible of the accuracy of a method, especially when the number of cases is small like the cases in this study.
The authors responded by saying that "Certainly, if they were comparing two groups, they should clearly understand the variability within and between those groups. However, in the present study, the groups from the two cohorts presented (RIH dataset, n=7; MDA dataset, n=66) cannot be directly compared quantitatively due to the small size of the RIH dataset. Instead, the presented data should be considered as a pilot dataset (RIH n=7), a larger retrospective dataset that qualitatively validates the first and provides a set of quantitative results that can be compared to a similarly sized dataset to be generated (or published) in the future. At that point, variability should be ascertained as in the cited literature."

Therefore, the authors are asked why, despite the recognition of the importance of this comment and of this suggested work, this very important answer has not been included in both the limitations of the study (since, as they say, "the data presented should be considered as a pilot dataset (RIH n=7), a larger retrospective dataset that qualitatively validates the first and provides a set of quantitative results that can be compared with a dataset of similar size to be generated (or published) in the future. "), but also in the conclusion. Furthermore, the authors are asked why, despite the affirmative answer to my comment, both the answer given to me and the citation of the work considered have not been included in the manuscript among the limitations and conclusions.

In light of the fact that radiomics and the cited work are highly influential on the state of the art, it is strongly necessary to take into account all the related and discussed state of the art, highlighting the advantages, limitations and future perspectives, in this direction, emphasizing that given the small dataset this is a pilot study and that in the future, as done in the mentioned studies, it will be applied to "a larger retrospective dataset that qualitatively validates the first one and provides a set of quantitative results that can be compared with a dataset of similar size".

Round 3

Reviewer 3 Report

Comments and Suggestions for Authors

The authors have correctly and meticulously responded to all comments and suggestions.
Now the manuscript is more explanatory and supported by considerable experimental evidence and can better capture the attention of future readers as the topic is of strong scientific impact. Thanks.